# Intense light as anticoagulant therapy in humans

Yoshimasa Oyama[1], Sydney Shuff[1], Pavel Davizon-Castillo[2], Nathan Clendenen[1], Tobias Eckle[1]*

1 Department of Anesthesiology, University of Colorado - Anschutz Medical Campus, Aurora, Colorado, United States of America, 2 Department of Pediatrics, University of Colorado - Anschutz Medical Campus, Aurora, Colorado, United States of America

* tobias.eckle@cuanschutz.edu

**Data Availability Statement:** Data are uploaded to figshare: https://doi.org/10.6084/m9.figshare.13424414.

**Funding:** The authors disclosed receipt of the following financial support for the research,

## Abstract

Blood coagulation is central to myocardial ischemia and reperfusion (IR) injury. Studies on the light elicited circadian rhythm protein Period 2 (PER2) using whole body *Per2*$^{-/-}$ mice found deficient platelet function and reduced clotting which would be expected to protect from myocardial IR-injury. In contrast, intense light induction of PER2 protected from myocardial IR-injury while *Per2* deficiency was detrimental. Based on these conflicting data, we sought to evaluate the role of platelet specific PER2 in coagulation and myocardial ischemia and reperfusion injury. We demonstrated that platelets from mice with tissue-specific deletion of *Per2* in the megakaryocyte lineage (*Per2*$^{loxP/loxP}$-PF4-CRE) significantly clot faster than platelets from control mice. We further found increases in infarct sizes or plasma troponin levels in *Per2*$^{loxP/loxP}$-PF4-CRE mice when compared to controls. As intense light increases PER2 protein in human tissues, we also performed translational studies and tested the effects of intense light therapy on coagulation in healthy human subjects. Our human studies revealed that intense light therapy repressed procoagulant pathways in human plasma samples and significantly reduced the clot rate. Based on these results we conclude that intense light elicited PER2 has an inhibitory function on platelet aggregation in mice. Further, we suggest intense light as a novel therapy to prevent or treat clotting in a clinical setting.

## Introduction

The role of platelets in the thrombotic occlusion of coronary vessels leading to myocardial ischemia is well understood [1–3]. Further, microembolization and platelet accumulation within the affected microcirculation of the myocardium during ischemia and reperfusion (IR) lead to secondary tissue damage [2, 3]. Interestingly, the early morning surge in blood pressure is accompanied by endothelial dysfunction, a peak in clinical thrombosis, and adverse cardiovascular events [4, 5]. These events correspond to oscillations in circadian gene and protein expressions, implicating a critical role of the circadian clock in these processes [6–9].

The circadian 'master' clock in mammals is in the hypothalamic suprachiasmatic nucleus (SCN). A hallmark of the mammalian circadian pacemaker is its ability to be synchronized by light [10]. Photic stimuli are transmitted from the retina to target neurons in the brain, where

authorship, and/or publication of this article: This work was supported by the National Heart, Lung, and Blood Institute (NIH-NHLBI) 5R01HL122472 Grant to TE and American Heart Association (AHA) Postdoctoral Fellowship 19POST34380105 to YO.

**Competing interests:** The authors have declared that no competing interests exist.

they are transduced to the molecular clockwork [11, 12]. Light activation of melanopsin receptors in the retinal ganglion cells leads to the transcriptional induction of Period 2 (PER2) and concomitant synchronization. Peripheral tissues display oscillations in PER2 expression similar to those of the brain [13, 14], probably through secreted signaling molecules [11, 15, 16]. Only light with an intensity >180 LUX can synchronize the human circadian system [17], where intense light (>10,000 LUX) is most effective.

Recently, the light-regulated circadian rhythm protein PER2 was identified as a critical endogenous protective mechanism to dampen the consequences of myocardial IR injury [14, 18–20]. Studies using tissue-specific mouse models for *Per2* identified a specific role for endothelial expressed PER2 in intense light elicited protection from myocardial IR injury [18]. Further, human studies found intense light elicited increases of PER2 protein and associated PER2-mechanisms [18].

Studies evaluating the role of PER2 for platelet function using PER2 whole body deficient mice found that PER2 deficiency was associated with a defective platelet function [21]. It was found that *Per2*$^{-/-}$ mice had prolonged bleeding times and platelet aggregation *in vitro* was significantly compromised in *Per2*$^{-/-}$ mice. As compromised platelet aggregation reduces myocardial IR-injury [22], these findings stand in contrast to recent findings on PER2 as an endogenous protective mechanism during myocardial ischemia [18]. Thus, we thought to investigate the tissue-specific role of PER2 for platelets. Using a recently described floxed mouse model for PER2 [18] we generated a tissue-specific mouse with a *Per2* deletion in the megakaryocyte lineage using the Platelet factor 4 (PF4) Cre recombinase mouse [23]. Using this novel mouse line, we evaluated platelet numbers, *in vitro* platelet aggregation, and myocardial ischemia and reperfusion injury *in vivo*. Also, we used intense light therapy which increases PER2 in human tissues and studied the effects of light on human plasma proteins and blood clotting. These studies reveal novel insights into the role of intense light elicited PER2 for blood clot formation.

## Material and methods

### Mouse experiments

Experimental protocols were approved by the Institutional Review Board (Institutional Animal Care and Use Committee [IACUC]) at the University of Colorado Denver, USA. They were following the AAALAC regulations, the US Department of Agriculture Animal Welfare Act, and the Guide for the Care and Use of Laboratory Animals of the NIH. All mice were housed in a 14 h (hours):10 h L(light):D(dark) cycle and we routinely used 12- to 16-week old male mice. To be able to compare new data with previous findings a LD14:10 light cycle was chosen which is the standard housing condition at the University of Colorado. Moreover, a LD14:10 cycle is commonly used and recommend by Jackson Laboratories to reduce mouse stress. All mice had a C57BL/6J background. C57BL/6J, *Per2*$^{-/-}$ [Per2tm1Brd Tyrc-Brd/J [24]], mice were purchased from Jackson laboratories. *Per2*$^{loxp/loxp}$ were generated by Ozgene (Perth, Australia) [18]. PF4-CRE [C57BL/6-Tg(Pf4-icre)Q3Rsko/J [23]], Lyz2-CRE [B6.129P2-Lyz2tm1(cre)Ifo/J [25]], tamoxifen-inducible Myosin-CRE [Tg(Myh6-cre/Esr1*)1Jmk/J [26]] or VE-Cadherin-Cre [B6.Cg-Tg(Cdh5-cre)7Mlia/J [27]] were purchased from Jackson laboratories. To obtain platelet, bone marrow, myocyte, or endothelial tissue-specific mice, we crossbred *Per2*$^{loxp/loxp}$ mice with the PF4-Cre, Lyz2-Cre, Myosin-Cre, or VE-Cadherin-Cre recombinase mouse. All mouse experiments were conducted at the same time point (ZT8, Zeitgeber Time corresponding to 2 PM based on 'light ON' at 6 AM) unless specified otherwise. Mice were bred in the vivarium at Denver for optimal acclimatization and housed in cages of 5 at 21°C with food (Harlan diets, formulation 2920x, soy-free) and water ad libitum. As studies have shown that

thrombotic vascular occlusion in mice is circadian controlled and significantly slowed down at Zeitgeber time (ZT) 8 [28], we chose this time point for our all studies on murine platelet aggregation.

## Platelet aggregation

Human blood was obtained by venipuncture into Vacutainer EDTA tube (BD, Franklin Lakes, NJ) from healthy volunteers. Blood sample from mice was drawn by cardiac puncture with sodium citrate. The platelet-rich plasma (PRP) was separated by centrifugation at 100 x g for 10 min, and platelet-poor plasma (PPP) was collected by centrifugation at 2000 x g for 20 min. Platelet counts in PRP were adjusted to $2.5 \times 10^5/\mu l$ with PPP. PRP was stirred (1200 rpm) at 37˚C in a Chrono-log model 700 (Chronolog, Havertown, PA). Aggregation was induced with 2μg/ml of collagen or 5μM adenosine diphosphate (ADP). Aggregation was recorded as the percent change in light transmission using Aggrolink software (Chronolog, Havertown, PA).

## Murine model for cardiac ischemia

Murine *in situ* myocardial ischemia and reperfusion injury (60-min ischemia/120 min reperfusion) and troponin-I (cTnI) measurements were performed as described [29–31]. Infarct sizes were determined by calculating the percentage of infarcted myocardium to the area at risk (AAR) using a double staining technique with Evan's blue and triphenyltetrazolium chloride. AAR and the infarct size were determined via planimetry using the NIH software Image 1.0 (National Institutes of Health, Bethesda, MA). For troponin I (cTnI) measurements blood was collected by central venous puncture and cTnI was analyzed using a quantitative rapid cTnI assay (Life Diagnostics, Inc., West Chester, PA, USA). **Note**: cTnI is highly specific for myocardial ischemia and has a well-documented correlation with the infarct size in mice [29, 31–33] and humans [34].

## Human light exposure

Based on strategies using intense light therapy [10,000 LUX] to treat seasonal mood disorders in humans [35], we adopted a similar protocol. Timing of light therapy has been shown to be critical as light therapy e.g. if given at night can cause insomnia and hyperactivity and therefore the morning hours are recommended. Healthy human volunteers were exposed to intense light (10,000 LUX) for 30 min every morning for five days from 8:30 AM– 9:00 AM. 5 mL blood was drawn on day one at 8:30 AM and 9:00 AM (before and after light exposure). While light exposure was repeated every morning for the five days, the next blood draws were on days three and five at 9:00 AM as indicated. Blood was collected in EDTA-plasma tubes and spun at 3,000 rpm for 8 minutes to separate plasma. We obtained approval from the Institutional Review Board (COMIRB #13–1607) for our human studies before written informed consent from everyone was obtained. A total of 6 healthy volunteers were enrolled (3 females and 3 males) [36].

## Proteomics screen

We analyzed plasma samples on day 1, day 3, and day 5 from healthy human volunteers exposed to 30 minutes of intense light in the morning on 5 consecutive days using the Slow Off-rate Aptamer (SOMAmer)-based capture array called SOMAscan [37, 38] (SomaLogic, Inc., CO, USA). The SOMAscan uses a protein signal present in the human plasma and transforms it into a nucleotide signal that can be quantified using fluorescence on microarrays. The

SOMAscan assay is one of the most comprehensive protein discovery tools available and measures 1319 plasma proteins (full list of light-regulated proteins published in [36]).

## Sonoclot coagulation analyzer

To measure clot formation and strength as well as the interaction between platelet and fibrin, a Sonoclot Coagulation Analyzer, a viscoelastic test instrument, was used with a glass bead test (Sienco® gbACT™ Kit) [39]. Citrated blood was re-calcified moments prior to the test assays to reverse the citrate's anticoagulant effect. An aliquot of 1 mL of citrated blood was mixed with 40 μL of CaCl$_2$ 0.25 M and then 330 μL of the re-calcified blood was added to the cuvette. A probe moves up and down along the vertical axis and as the sample starts to clot, changes in impedance to movement are measured. The time-based graph (Sonoclot signature) that is generated reflects different steps in the clotting of the whole blood sample in three different variables. Test variability of the Sonoclot analyses was determined to be 6–10%. The following variables were measured, with the defined normal values in parentheses: Activated clotting time (ACT) (100–155 s) is the time required for the first fibrin to form. Clot rate (CR) (9–35 units/min) is the rate of increase in the clot impedance due to fibrin formation and polymerization [39].

## Data analysis

Two groups comparisons were analyzed using a Student's t-test. The protein array data were analyzed via linear regression with false discovery rate correction of the p values using the Benjamini and Hochberg method to control for multiple comparisons. Values are expressed as mean (±SD). $P < 0.05$ was considered statistically significant. The Kolmogorov Smirnov test was used to confirm normal distribution. For statistical analysis, GraphPad Prism 7.0 or JMP 14 software were used.

## Results

### Platelet counts in $Per2^{loxP/loxP}$-PF4-CRE mice are unchanged

Studies using whole-body $Per2$ deficient mice found increased bleeding time, compromised platelet function, and reduced total platelet counts [21]. To further understand the role of PER2 for platelet function we used a novel floxed mouse model for PER2 [18] (Fig 1) and crossbred $Per2^{loxP/loxP}$ with the platelet factor 4 (PF4)-CRE mouse [23] to obtain a tissue-specific deletion of $Per2$ exclusively in the megakaryocytic lineage (Fig 1). As studies in $Per2$ whole-body knockout mice found reduced platelet counts when compared to controls, we first analyzed platelet numbers and mean platelet volume in blood samples from $Per2^{loxP/loxP}$-PF4-CRE and PF4-CRE mice. As shown in Fig 1, platelet counts, or mean platelet volume did not differ between $Per2^{loxP/loxP}$-PF4-CRE and PF4-CRE controls. However, $Per2^{-/-}$ had indeed lower platelet numbers as seen in previous studies. Together, tissue-specific deletion of $Per2$ in the megakaryocytic lineage does not change blood platelet counts.

### Increased platelet aggregation in $Per2^{loxP/loxP}$-PF4-CRE mice

We next isolated platelets from $Per2^{loxP/loxP}$-PF4-CRE and PF4-CRE controls and analyzed ADP mediated platelet aggregation using an aggregometer. As studies have shown that thrombotic vascular occlusion in mice is circadian controlled and significantly slowed down at Zeitgeber time (ZT) 8, which is abolished in $Clock$ mutant mice [28], we chose this time point for our following studies. We demonstrated that ADP mediated platelet aggregation from isolated $Per2^{loxP/loxP}$-PF4-CRE-platelets was significantly enhanced at ZT8 when compared to platelets

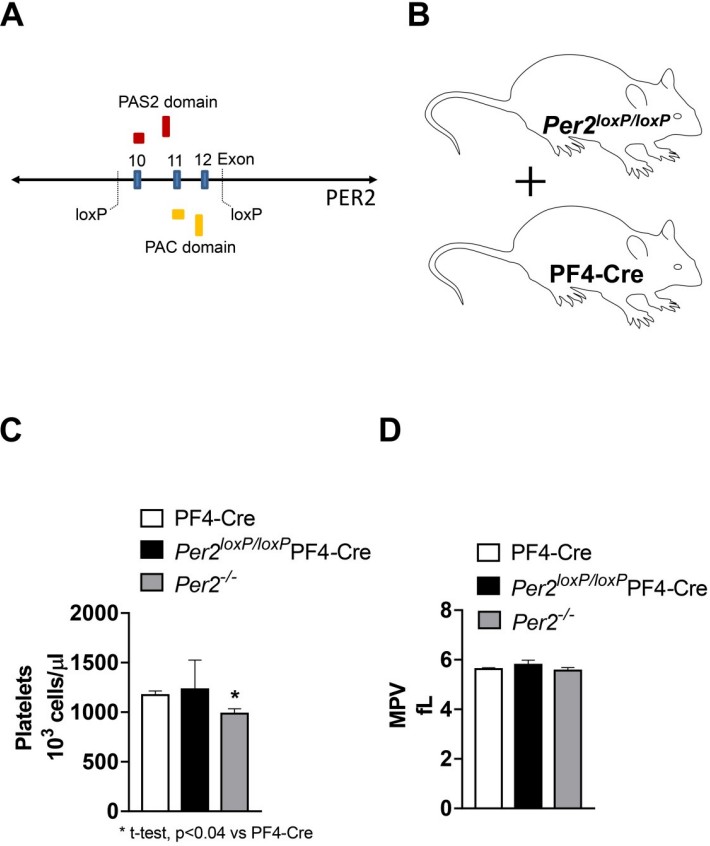

**Fig 1. Platelet counts in *Per2^loxP/loxP*-PF4-CRE mice.** (**A**) *Per2^loxp/loxp* -strategy: deletion of exons 10, 11 and 12 in the Per2 gene removes half of the PAS2 domain and all the PAC domain. This deletion also results in a frameshift mutation introducing an early stop codon. (**B**) Generation of mice with a tissue-specific deletion of Per2 exclusively in the megakaryocytic lineage. (**C**) Platelet numbers and mean platelet volume 9MPV) from PF4-CRE, *Per2^loxP/loxP*-PF4-CRE and *Per2^-/-* mice (mean±SD, n = 3–4).

isolated from PF4-CRE controls (Fig 2). Similarly, collagen-induced platelet aggregation was also significantly increased in platelets from *Per2^loxP/loxP*-PF4-CRE mice at ZT8 (Fig 2). Together, platelet aggregation is significantly enhanced in platelets isolated from mice with tissue-specific deletion of *Per2* in the megakaryocytic lineage at ZT8.

## Increased myocardial damage in *Per2^loxP/loxP*-PF4-CRE mice

After we found enhanced platelet clot formation at ZT8 in *Per2^loxP/loxP*-PF4-CRE, we next exposed *Per2^loxP/loxP*-PF4-CRE and PF4-CRE controls to *in-situ* myocardial ischemia and reperfusion injury at ZT8. We exposed mice to 60 minutes of myocardial ischemia and 2 hours of reperfusion using a hanging weight system [30]. Following reperfusion, mouse cardiac tissue was stained using triphenyltetrazolium chloride (TTC). As shown in Fig 3, *Per2^loxP/loxP*-PF4-CRE had lager infarct sizes than PF4-CRE controls, which was however not significant (54%±7% vs 45%±7%, mean±SD, p = 0.1087). Similarly, plasma troponin levels were also -but not significantly- increased in *Per2^loxP/loxP*-PF4-CRE mice (204±94 ng/ml vs 110±67 ng/ml, mean±SD, p = 0.1). Together, while not significant, *Per2^loxP/loxP*-PF4-CRE mice have larger infarct sizes and serum troponin levels following myocardial ischemia and reperfusion injury at ZT8 when compared to controls.

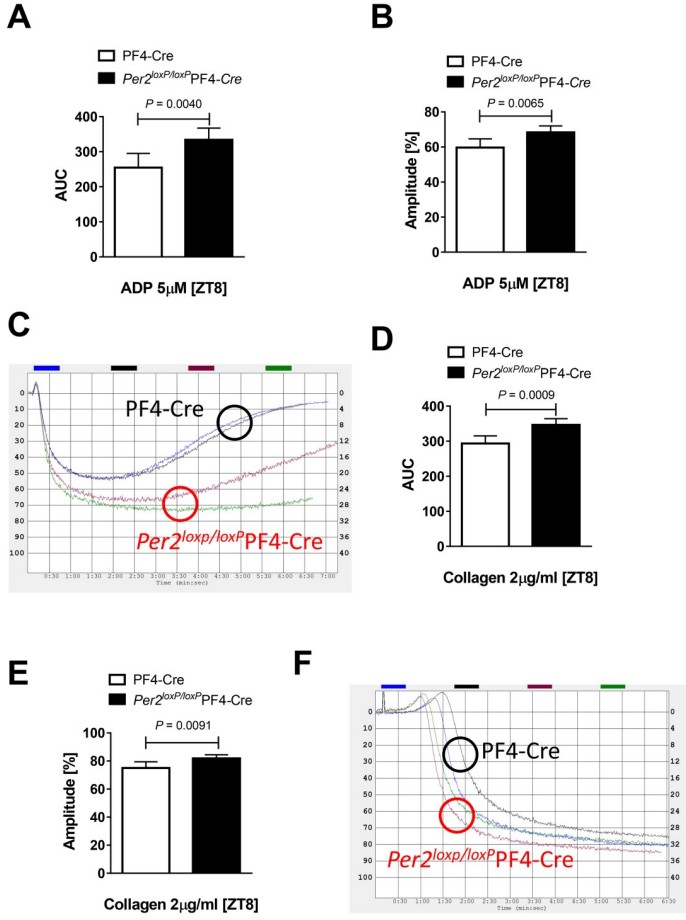

**Fig 2. Platelet aggregation in *Per2^{loxP/loxP}*-PF4-CRE mice.** Platelet-rich plasma (PRP) was obtained from *Per2^{loxP/loxP}*-PF4-CRE and PF4-CRE controls. Aggregation was induced with 5μM adenosine diphosphate (ADP) (**A-C**) or 2μg/ml of collagen (**D-F**) and recorded as the percent change in light transmission using an aggregometer (mean±SD; n = 6).

### Intense light therapy inhibits procoagulant pathways in humans

Based on reports that humans clot more in the early morning than in the afternoon [40], we next compared platelet aggregation in humans at 9 AM versus 4 PM. Indeed, as shown in Fig 4, platelet aggregation was enhanced at 9 AM vs 4 PM in healthy human subjects. As 9 AM appeared to be the time point where therapy would be desirable, we next evaluated intense light therapy as a strategy to reduce coagulation in healthy human subjects at 9 AM (Fig 4). Recent studies demonstrated that 5 days of intense light therapy enhances PER2 protein expression in tissue samples from healthy human subjects at 9 AM [18]. Using the SomaScan protein array, we analyzed intense light-regulated plasma proteins following 5 days of 30 minutes intense light therapy between 8.30–9.00 AM [36]. Several pathway analyses, as shown in Fig 4, indicated that intense light significantly inhibited procoagulant pathways. As shown in Fig 5, volcano plot analysis of our proteomics screen confirmed the findings of our pathway analyses and suggested a strong regulation of platelet factor 4 (PF4) by intense light. Taken together, intense light therapy, which increases PER2 in humans, inhibits procoagulant proteins in human plasma samples.

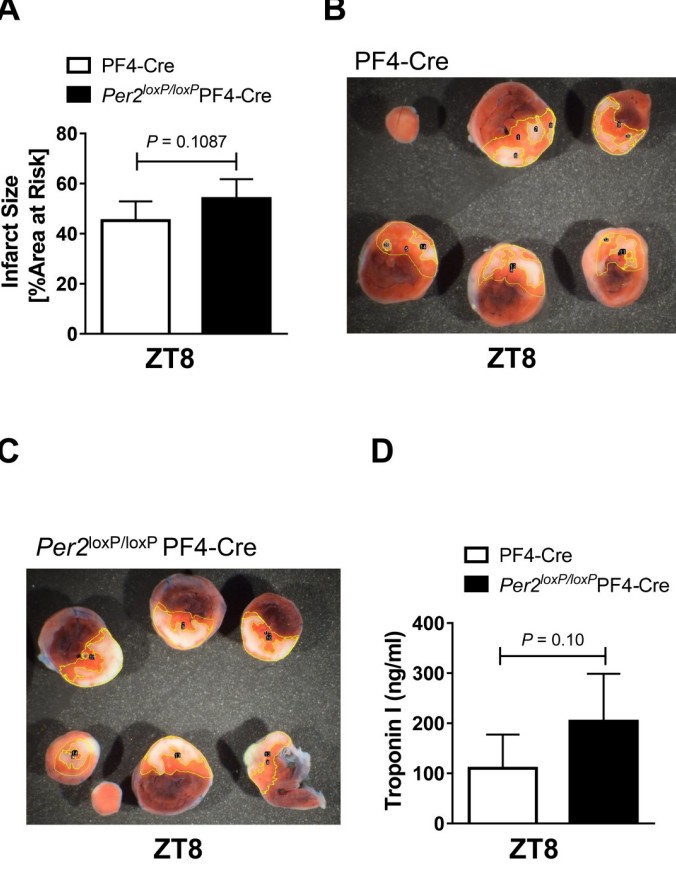

**Fig 3. Myocardial ischemia and reperfusion injury in *Per2^loxP/loxP*-PF4-CRE mice.** Infarct sizes (**A**) or serum troponin-I (**D**) in *Per2^loxP/loxP*-PF4-CRE and PF4-CRE controls mice after 60 min of in situ myocardial ischemia and 2h reperfusion at ZT8. (**B-C**) Representative infarct staining (mean±SD; n = 5).

## Intense light therapy inhibits clot rate in humans

After we found that intense light inhibited procoagulant proteins in human plasma samples, we next tested the effect of intense light on whole blood coagulation using the Sonoclot. As shown in Fig 6, 30 minutes of intense light therapy on day one demonstrated an immediate effect on activated clotting time (initial phase, the time required for the first fibrin to form, non-platelet dependent) and clot rate (second phase, the rate of increase in the clot impedance, platelet dependent). After 5 days of intense light therapy, the clot rate was further significantly reduced when compared to post light treatment on day 1. However, activated clotting time on day 5 returned to baseline values from day 1 (Fig 6). Together, 5 days of intense light therapy significantly reduces the clot rate in healthy human subjects.

## Discussion

Acute coronary thrombosis can result in nonfatal myocardial infarction or sudden death [41]. This process is well defined in patients with heart failure, patients with coronary artery disease, and those dying of sudden cardiac death. Circadian mechanisms regarding thrombosis have been reported but are not well defined due to the lack of tissue-specific studies [24, 42]. In the current study we observed that tissue-specific deletion of the circadian and light-regulated

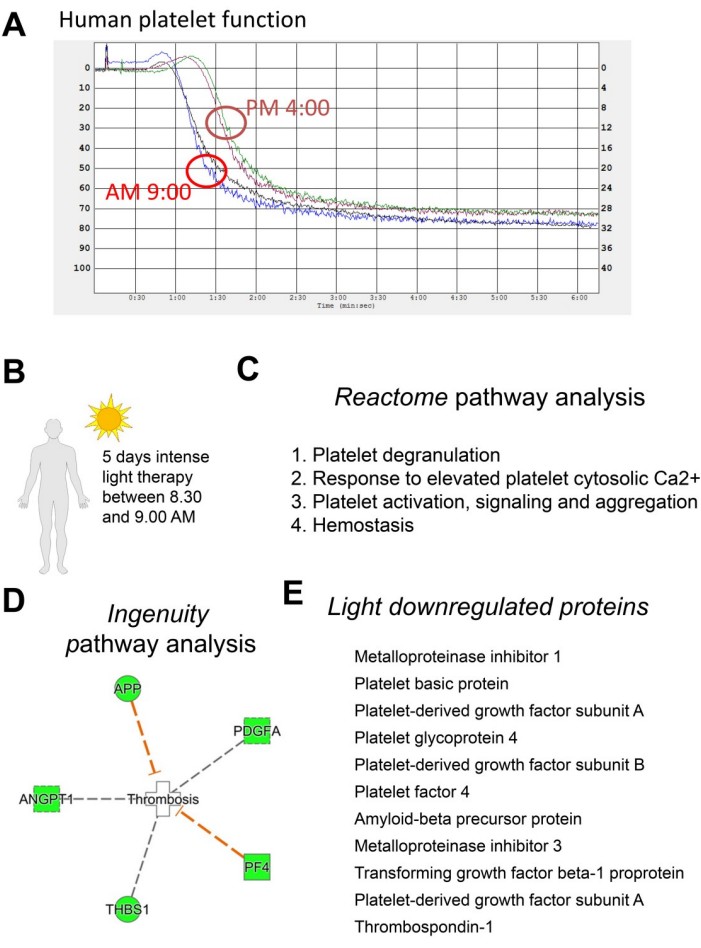

**Fig 4. Proteomics from human plasma samples after intense light therapy.** (**A**) Platelet-rich plasma (PRP) was obtained from human healthy volunteers. Aggregation was induced with 1 μl/ml collagen and recorded as the percent change in light transmission using an aggregometer. (**B**) Healthy human volunteers were exposed to 30 minutes of intense light (10,000 Lux) at 8:30 AM on 5 consecutive days. A blood draw was performed before light exposure on the first day (8:30 AM) and 3 or 5 days after light exposure (9.00 AM). Plasma samples were analyzed using the SOMAscan platform. (**C**) Reactome analysis of intense light-regulated proteins. (**D**) Ingenuity analysis of intense light-regulated proteins. (**E**) All light-regulated proteins affecting coagulation (n = 3 individual subjects, n = 12 of total samples/arrays).

protein PER2 in the megakaryocyte lineage results in increased platelet aggregation and increased myocardial damage. Further, we demonstrated that intense light therapy -which increases PER2 protein tissue levels- inhibited procoagulant pathways and reduced the clot rate in healthy human subjects.

Mechanistic studies on how circadian proteins influence coagulation are scarce. Moreover, most studies have used whole body knockout mouse models evaluating the circadian clock in hemostasis. As such, a study using whole-body Per2 knockout mice found that *Per2*-null mice had reduced platelet counts and platelets were compromised in their ability to aggregate [21]. Furthermore, an ultrastructural examination of *Per2*-null megakaryocytes revealed many vacuoles in demarcation membranes and a reduction in platelet granules [21].

In a different study, it was found that there was a diurnal rhythm in the expression of thrombopoietin in wildtype mice while in *Clock* mutant mice thrombopoietin expression was disrupted [42]. In contrast to the study using whole body *Per2*$^{-/-}$ mice, however, *Clock* mutant

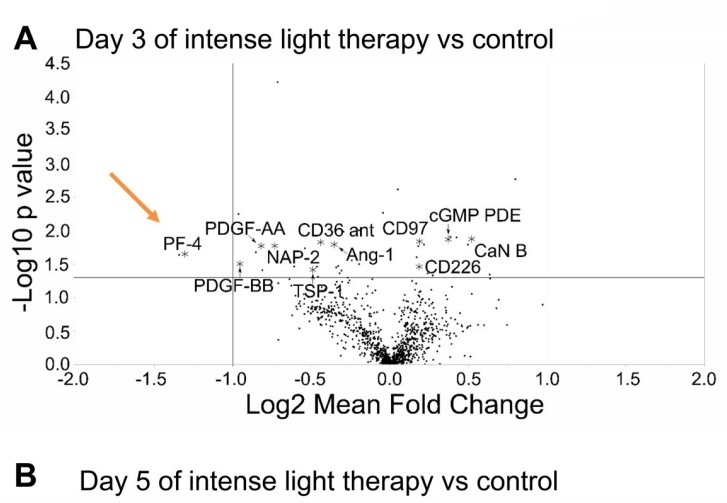

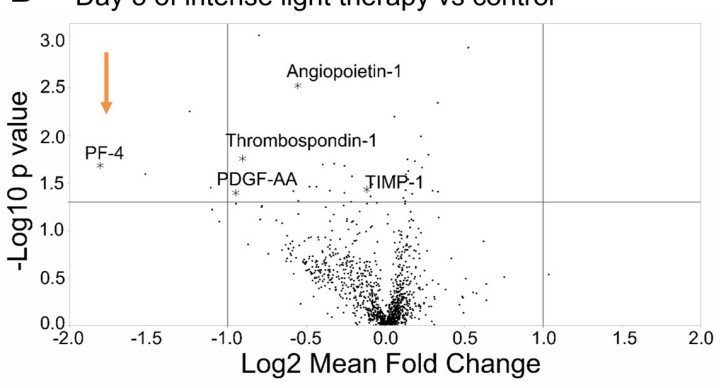

**Fig 5. Volcano plot analysis of a proteomic screen from human plasma sample following intense light therapy.**
Healthy human volunteers were exposed to 30 minutes of intense light (10,000 Lux) at 8:30 AM on 5 consecutive days.
A blood draw was performed before light exposure on the first day (8:30 AM) and 3 or 5 days after light exposure (9.00
AM). Plasma samples were analyzed using the SOMAscan platform. Shown are volcano pot analyses of the all light
regulated proteins. Red arrows mark the strong effect of intense light on platelet factor 4 (PF4; n = 3 individual
subjects, n = 12 of total samples/arrays).

mice showed an increase in thrombopoietin, a significant increase in megakaryocyte numbers
and significant higher platelet counts at ZT8. Unfortunately, no analysis of the platelet func-
tion was performed.

In our current study using mice with a *Per2* deletion in the megakaryocyte lineage, we did
not see changes in platelet counts but found increased platelet aggregation at ZT8. We chose
this time point as wildtype mice have prolonged thrombotic vascular occlusion times (reduced
clotting) at ZT8 [28]. Indeed, *Clock* mutant mice show abolished diurnal variation of throm-
botic vascular occlusion with enhanced *in vivo* thrombosis at ZT8 [28].

Another study using whole-body *Bmal1*$^{-/-}$ found enhanced platelet aggregation upon ADP
stimulation [43]. Surprisingly, disruption of endothelial BMAL1 expression was found to sig-
nificantly shorten thrombotic vascular occlusion at ZT8 [28]. These data indicate that different
clock proteins control different functions in different tissues. Further, these data show that
results from whole-body null mice affecting circadian core clock proteins are difficult to inter-
pret. Nevertheless, all these studies highlight a critical role of the circadian system in thrombo-
sis generation.

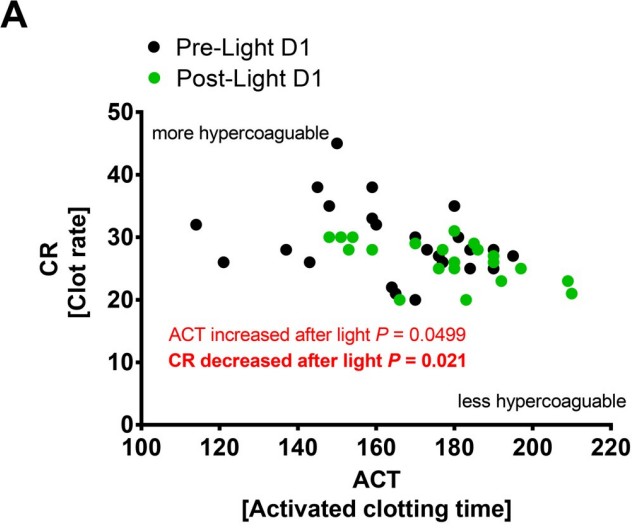

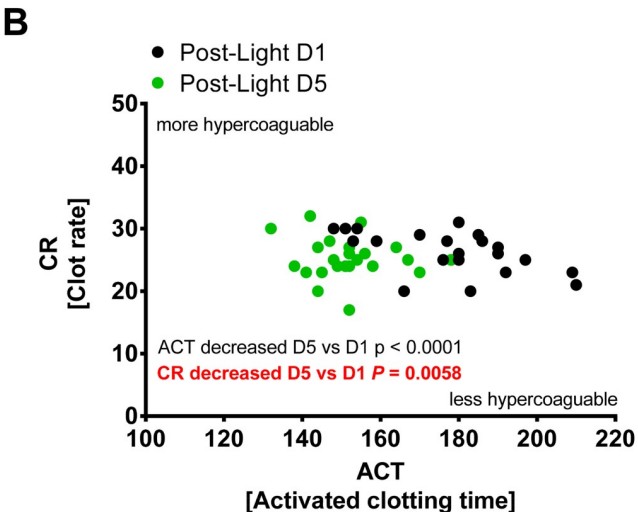

**Fig 6. Whole blood coagulation studies in human subjects during intense light therapy.** Healthy human volunteers were exposed to 30 minutes of intense light (10,000 Lux) at 8:30 AM on 5 consecutive days. A blood draw was performed before light exposure on the first day (8:30 AM) and 3 or 5 days after light exposure (9.00 AM). Activated clotting time (ACT) and Clot rate (CR) were determined from whole blood using the Sonoclot Coagulation Analyzer (mean±SD; n = 6 individual subjects).

The contribution of platelet activation to myocardial ischemia and reperfusion injury has been well documented [1]. Transfusion of myocardial ischemia-activated platelets from wild-type into wildtype mice resulted in increased myocardial damage [1]. Possible mechanisms include upregulation of platelet surface receptors and release of immunomodulatory mediators, microembolization or modification of the cardiac vascular endothelium, which all can lead to aggravation of myocardial ischemia and reperfusion injury [44]. While in the current studies we found that a tissue-specific deletion of the circadian and light-regulated protein PER2 in the megakaryocyte lineage results in increased platelet aggregation, the myocardial damage, however, was moderate. In fact, this is in line with recent studies that found endothelial expressed PER2 as the dominant player in protection from myocardial ischemia and reperfusion injury [18]. On the other side, PER2 appears to play an important role during

myocardial ischemia and reperfusion injury in general, as tissue-specific deletion of *Per2* in myocytes, bone marrow cells, megakaryocytes or endothelial cells increase myocardial damage (Fig 3, S1 Fig). Future studies would have to test the interaction and the importance of PER2 in different tissues and analyze thrombotic vascular occlusion in mice that are deficient in endothelial PER2.

The importance of light as a regulator of the circadian system has been well described [8, 9, 16, 18, 20]. Our group recently found that intense light provides robust protection from myocardial ischemia and reperfusion injury [18]. These studies identified endothelial expressed *Per2* as a critical component of intense light-mediated cardioprotection. Interestingly, studies on platelet turnover found that megakaryopoiesis is regulated by light signals emanating from the master oscillator within the SCN of the hypothalamus [45]. Our studies have shown that light increases PER2 levels in peripheral tissues in mice and humans [18]. Based on these observations we evaluated intense light as a therapy to increase PER2 and to possibly affect coagulation. We found that intense light creates an anti-thrombotic signature in plasma samples and that the clot rate, which is platelet dependent, is significantly reduced after 5 days of intense light therapy in human healthy subjects. Despite our first promising results from healthy human subjects, further research will be necessary to understand the mechanisms of intense light in inhibiting coagulation fully.

While mice are nocturnal and humans are diurnal, our group has shown that myocardial ischemia leads to bigger infarct sizes and plasma troponin values in the early morning hours like studies in humans [46]. In fact, circadian rhythms function independently of a diurnal or nocturnal behavior due to multiple yet parallel outputs from the SCN [47]. As such, very basic features of the circadian system are the same in apparently diurnal and nocturnal animals, including the molecular oscillatory machinery and the mechanisms responsible for pacemaker entrainment by light [47]. Further, we have shown that PER2 levels reciprocally correlate with infarct sizes and troponin levels in mice and men [14, 18]. These findings highlight that activity levels do not affect the diurnal pattern of myocardial ischemia events, as proposed by some authors that the increased morning incidence of myocardial ischemia in humans is purely stress related [48]. Also, PER2 is hypoxia-regulated in mice and humans, which supports similar biological roles in both species [14]. Further, important hypoxia adaptive mechanism such as hypoxia inducible factor 1 (HIF1A) regulation and function are PER2 dependent [49] and also seem to be independent of a nocturnal nature [50], despite HIF1A expression being under circadian control [51]. Indeed, human and mouse studies on HIF1A find similar responses to cardiovascular ischemic events [50].

Our data are not without limitations and should be interpreted with caution. While we found an anti-thrombotic effect of intense light elicited PER2 or intense light in mice or humans, respectively, differences in size and physiology, as well as variations in the homology of targets between mice and humans, may lead to translational limitations. Further limitations of our work are small sample sizes, only using one time point for mice or human studies and not evaluating clotting events in humans in the evening following intense light exposure in the morning. Moreover, the anti-thrombotic signature observed in human plasma might not reflect the real coagulation status. Besides, low sample size and test limitations (selection of 1319 proteins) of our proteomics platform might make our conclusions on light having an impact on humans appear premature. Nevertheless, we have analyzed 12 plasma samples from 4 healthy volunteers over a week in our SomaScan assay. For our clotting studies using the Sonoclot, we analyzed whole blood from 6 healthy subjects over one week. While intense light therapy has been validated for the treatment of seasonal disorders, studies on the biological effects of intense light are scarce. In fact, to our knowledge, there are no studies that have performed a wide protein screen from plasma samples following intense light therapy in humans.

The SOMAscan platform, which we chose, is a highly multiplexed, aptamer-based assay optimized for protein biomarker discovery, which is made possible by the simultaneous measurement of a broad range of protein targets. This assay has been successful in the identification of biomarker signatures in a variety of recent biomedical applications [52]. As such, despite the limitations of our analysis, our results will hopefully stimulate future research on the role of intense light therapy in the regulation of pro or anti-coagulant processes.

## Conclusions

We have demonstrated that tissue-specific deletion of *Per2* in the megakaryocyte lineage increases platelet aggregation and thus, PER2 could represent a novel drug target to treat procoagulant disease states. Further, our human studies indicate that intense light -which is the dominant regulator of circadian rhythms and PER2- could potentially be a novel therapy to prevent or treat blood clotting in a clinical setting.

## Supporting information

**S1 Fig. Myocardial ischemia in mice with a tissue specific deletion in endothelia, myocytes or bone marrow.** Serum troponin-I from *Per2*$^{loxP/loxP}$-VE Cadherin Cre (endothelial specific), *Per2*$^{loxP/loxP}$-Myosin Cre (cardiomyocyte sepcfic), *Per2*$^{loxP/loxP}$-Lyz2 Cre (bone marrow specific) after 60 min of in situ myocardial ischemia and 2h reperfusion (mean±SD; n = 5). (PDF)

## Author Contributions

**Conceptualization:** Tobias Eckle.

**Data curation:** Yoshimasa Oyama, Pavel Davizon-Castillo, Tobias Eckle.

**Formal analysis:** Yoshimasa Oyama, Pavel Davizon-Castillo, Nathan Clendenen, Tobias Eckle.

**Funding acquisition:** Yoshimasa Oyama, Tobias Eckle.

**Investigation:** Yoshimasa Oyama, Pavel Davizon-Castillo, Tobias Eckle.

**Methodology:** Yoshimasa Oyama, Nathan Clendenen, Tobias Eckle.

**Project administration:** Tobias Eckle.

**Supervision:** Tobias Eckle.

**Validation:** Sydney Shuff, Pavel Davizon-Castillo, Nathan Clendenen, Tobias Eckle.

**Visualization:** Yoshimasa Oyama, Sydney Shuff, Pavel Davizon-Castillo, Nathan Clendenen, Tobias Eckle.

**Writing – original draft:** Yoshimasa Oyama, Sydney Shuff, Pavel Davizon-Castillo, Nathan Clendenen, Tobias Eckle.

**Writing – review & editing:** Yoshimasa Oyama, Sydney Shuff, Pavel Davizon-Castillo, Nathan Clendenen, Tobias Eckle.

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
