## [Decision Letter · Decision Letter 0]

4 Nov 2020

PONE-D-20-20729

Intense light as anticoagulant therapy in humans

PLOS ONE

Dear Dr. Eckle,

Thank you for submitting your manuscript to PLOS ONE. After careful consideration, we feel that it has merit but does not fully meet PLOS ONE’s publication criteria as it currently stands. Therefore, we invite you to submit a revised version of the manuscript that addresses the points raised during the review process.

In particular, it is important that you note the sample sizes used for all experiments and how the small sample sizes for humans may limit your findings. Additionally, it would be important for you to comment on the particular phasing of the normal cardiovascular cycles for the diurnal humans and nocturnal mice, as has been denoted in the reviews. Similarly, you should also denote the limitations of only having samples from one phase of the circadian cycle.

We look forward to receiving your revised manuscript.

Kind regards,

Paul A. Bartell

Academic Editor

PLOS ONE

Journal Requirements:

2.In your Data Availability statement, you have not specified where the minimal data set underlying the results described in your manuscript can be found. PLOS defines a study's minimal data set as the underlying data used to reach the conclusions drawn in the manuscript and any additional data required to replicate the reported study findings in their entirety. All PLOS journals require that the minimal data set be made fully available. For more information about our data policy, please see http://journals.plos.org/plosone/s/data-availability.

Reviewers' comments:

Reviewer's Responses to Questions

**Comments to the Author**

1. Is the manuscript technically sound, and do the data support the conclusions?

Reviewer #1: Yes

Reviewer #2: Partly

2. Has the statistical analysis been performed appropriately and rigorously? 

Reviewer #1: Yes

Reviewer #2: No

3. Have the authors made all data underlying the findings in their manuscript fully available?

Reviewer #1: Yes

Reviewer #2: Yes

4. Is the manuscript presented in an intelligible fashion and written in standard English?

Reviewer #1: Yes

Reviewer #2: Yes

5. Review Comments to the Author

Reviewer #1: 1. The first part of the Abstract (and Introduction) needs clarification. Why are statements made in the first two sentences regarded as representing “conflicting data”? On the one hand, mice lacking PER2 have deficient platelet function (which can be expected since the circadian clock is impaired). On the other hand, intense light induction of PER2 protected from myocardial IR-injury (which can also be expected since the circadian system is boosted). In what way are these results contradictory?

2. All experiments in mice were carried out at ZT8, or 8 hours after light onset on an L14:D10 schedule.

a. Why was an L14D10 schedule chosen rather than L10D14 since mice are nocturnally active? ZT8 would roughly correspond to mid-rest.

b. Under Results, it is explained that ZT8 was chosen because thrombotic vascular occlusion is slowed down in mice at that time. It would help to also have some additional information for the choice of both ZT8 and L14D10 in the Methods section.

c. One question related to this choice, however, is that in humans, the incidence of myocardial infarctions peaks during the first few hours after awakening, not in the middle of the night. Addressing this issue in the Discussion may be helpful.

3. All variables investigated in this study are circadian periodic. Testing at a single circadian stage (in both mice and humans) is likely to have yielded only partial information. Repeating the measurements at different circadian stages would have been ideal. Mentioning this limitation in the Discussion is recommended.

4. Humans were exposed to light in the morning. Blood sampling was also done in the morning.

a. Would similar results be obtained should blood be sampled later in the day or during the night? Not all myocardial infarctions occur in the morning.

b. Adding information why exposure to light is preferred in the morning would be helpful (light at night is associated with circadian disruption and is usually seen as being harmful).

Reviewer #2: This article is very interesting and significant. In my opinion the study was planned very well.

I have some doubts about:

- small numer of human volunteers – too small?

- number of mice used in experiments - there is no information?

- tests which were used to perform a statistical analysis - a small number of healthy human volunteers

This requires some explanation.

6. PLOS authors have the option to publish the peer review history of their article (what does this mean?). If published, this will include your full peer review and any attached files.

Reviewer #1: **Yes: **GERMAINE CORNELISSEN

Reviewer #2: No

---

## [Author Response · Author response to Decision Letter 0]

11 Nov 2020

Point by point response

Editor

1. It is important that you note the sample sizes used for all experiments and how the small sample sizes for humans may limit your findings. 

Thank you so much for pointing this out. We agree that the small sample size for humans limits our findings. We have added this limitation to the discussion. All n numbers are added to the figure legends.

‘Further limitations of our work are small sample sizes, only using one time point for mice or human studies and not evaluating clotting events in humans in the evening following intense light exposure in the morning’

‘Besides, low sample size and test limitations (selection of 1319 proteins) of our proteomics platform might make our conclusions on light having an impact on humans appear premature.’

2. It would be important for you to comment on the particular phasing of the normal cardiovascular cycles for the diurnal humans and nocturnal mice, as has been denoted in the reviews. 

We agree that this is an important point. Thus, we have expanded on this in more detail in the discussion. In fact, we and others have found similar timing of events in humans and mice. As such humans have larger infarcts and troponin levels in the morning hours and we have shown in several studies that this is true for mice as well (e.g. Nature Medicine 2012, Cell Reports 2019). In fact, very fundamental features of the circadian system are the same in apparently diurnal and nocturnal animals, including the molecular oscillatory machinery and the mechanisms responsible for pacemaker entrainment by light. As such, circadian rhythms function independently of a diurnal or nocturnal behavior due to multiple yet parallel outputs from the SCN (suprachiasmatic nuclei). 

‘While mice are nocturnal and humans are diurnal, our group has shown that myocardial ischemia leads to bigger infarct sizes and plasma troponin values in the early morning hours like studies in humans [46]. In fact, circadian rhythms function independently of a diurnal or nocturnal behavior due to multiple yet parallel outputs from the SCN [47]. As such, very basic features of the circadian system are the same in apparently diurnal and nocturnal animals, including the molecular oscillatory machinery and the mechanisms responsible for pacemaker entrainment by light [47]. Further, we have shown that PER2 levels reciprocally correlate with infarct sizes and troponin levels in mice and men [14, 18]. These findings highlight that activity levels do not affect the diurnal pattern of myocardial ischemia events, as proposed by some authors that the increased morning incidence of myocardial ischemia in humans is purely stress related [48]. Also, PER2 is hypoxia-regulated in mice and humans, which supports similar biological roles in both species [14]. Further, important hypoxia adaptive mechanism such as hypoxia inducible factor 1 (HIF1A) regulation and function are PER2 dependent [49] and also seem to be independent of a nocturnal nature [50], despite HIF1A expression being under circadian control [51]. Indeed, human and mouse studies on HIF1A find similar responses to cardiovascular ischemic events [50].’

3. You should also denote the limitations of only having samples from one phase of the circadian cycle.

Thank you for pointing this out. We agree that using only one phase limits our data and we added this to the discussion section.

‘Further limitations of our work are small sample sizes, only using one time point for mice or human studies and not evaluating clotting events in humans in the evening following intense light exposure in the morning.’

Reviewer #1 

1. The first part of the Abstract (and Introduction) needs clarification. Why are statements made in the first two sentences regarded as representing “conflicting data”? On the one hand, mice lacking PER2 have deficient platelet function (which can be expected since the circadian clock is impaired). On the other hand, intense light induction of PER2 protected from myocardial IR-injury (which can also be expected since the circadian system is boosted). In what way are these results contradictory?

Thank you so much for this excellent point. We apologize for not being clearer. The conflicting results are that PER2 KO mice clot less than wildtype mice. As less clotting would be associated with a benefit during myocardial ischemia and reperfusion injury, those findings conflict with findings showing larger infarct sizes following myocardial ischemia in PER2KO mice. We have clarified this in the abstract and the introduction.

‘Blood coagulation is central to myocardial ischemia and reperfusion (IR) injury. Studies on the light elicited circadian rhythm protein Period 2 (PER2) using whole body Per2-/- mice found deficient platelet function and reduced clotting which would be expected to protect from myocardial IR-injury. In contrast, intense light induction of PER2 protected from myocardial IR-injury while Per2 deficiency was detrimental. Based on these conflicting data, we sought to evaluate the role of platelet specific PER2 in coagulation and myocardial ischemia and reperfusion injury.’

‘As compromised platelet aggregation reduces myocardial IR-injury [22], these findings stand in contrast to recent findings on PER2 as an endogenous protective mechanism during myocardial ischemia [18].’

2. All experiments in mice were carried out at ZT8, or 8 hours after light onset on an L14:D10 schedule.

a. Why was an L14D10 schedule chosen rather than L10D14 since mice are nocturnally active? ZT8 would roughly correspond to mid-rest.

Thank you so much for pointing this out. The reason for this time schedule was that all our studies in mice have been conducted using a L10D14 light cycle. To be able to compare the new data with our own findings we felt it was critical to keep this light cycle. In addition, 14-hour light/10-hour dark cycle is a commonly used light cycle in animal facilities and recommended by Jackson laboratories to reduce mouse stress.

b. Under Results, it is explained that ZT8 was chosen because thrombotic vascular occlusion is slowed down in mice at that time. It would help to also have some additional information for the choice of both ZT8 and L14D10 in the Methods section.

Thank you so much for this excellent point. We have further expanded on his in the method section.

‘To be able to compare new data with previous findings a LD14:10 light cycle was chosen which is the standard housing condition at the University of Colorado. Moreover, a LD14:10 cycle is commonly used and recommend by Jackson Laboratories to reduce mouse stress.’

‘As studies have shown that thrombotic vascular occlusion in mice is circadian controlled and significantly slowed down at Zeitgeber time (ZT) 8 [28], we chose this time point for our all studies on murine platelet aggregation.’

c. One question related to this choice, however, is that in humans, the incidence of myocardial infarctions peaks during the first few hours after awakening, not in the middle of the night. Addressing this issue in the Discussion may be helpful.

Thank you for this excellent point. Our data have shown that MIs and troponin values peak in the early morning hours in mice and men and correlate with PER2 levels. We have expanded on this in the discussion.

‘While mice are nocturnal and humans are diurnal, our group has shown that myocardial ischemia leads to bigger infarct sizes and plasma troponin values in the early morning hours like studies in humans [46]. In fact, circadian rhythms function independently of a diurnal or nocturnal behavior due to multiple yet parallel outputs from the SCN [47]. As such, very basic features of the circadian system are the same in apparently diurnal and nocturnal animals, including the molecular oscillatory machinery and the mechanisms responsible for pacemaker entrainment by light [47]. Further, we have shown that PER2 levels reciprocally correlate with infarct sizes and troponin levels in mice and men [14, 18]. These findings highlight that activity levels do not affect the diurnal pattern of myocardial ischemia events, as proposed by some authors that the increased morning incidence of myocardial ischemia in humans is purely stress related [48]. Also, PER2 is hypoxia-regulated in mice and humans, which supports similar mechanisms in both species [14]. Further, important hypoxia adaptive mechanism such as hypoxia inducible factor 1 (HIF1A) regulation and function are PER2 dependent [49] and also seem to be independent of a nocturnal nature [50], despite HIF1A expression being under circadian control [51]. Indeed, human and mouse studies on HIF1A find similar responses to cardiovascular ischemic events [50].’

3. All variables investigated in this study are circadian periodic. Testing at a single circadian stage (in both mice and humans) is likely to have yielded only partial information. Repeating the measurements at different circadian stages would have been ideal. Mentioning this limitation in the Discussion is recommended.

We fully agree and added this as limitation to the discussion section. 

‘Further limitations of our work are small sample sizes, only using one time point for mice or human studies and not evaluating clotting events in humans in the evening following intense light exposure in the morning.’

4. Humans were exposed to light in the morning. Blood sampling was also done in the morning.

a. Would similar results be obtained should blood be sampled later in the day or during the night? Not all myocardial infarctions occur in the morning.

This is an excellent point. We don’t know the answer to this as we are at the beginning to understand how light therapy can affect important physiological processes. Since our data are so new, we felt that it would be important to share the data with the research community. Regarding light exposure and PER2 levels in humans, we have published that the short light exposure increases PER2 in humans in the morning and in the evening (way after any light exposure). So, we could speculate that this would also be the case for the anticoagulatory effects. However, these clearly needs further evaluation.

b. Adding information why exposure to light is preferred in the morning would be helpful (light at night is associated with circadian disruption and is usually seen as being harmful).

Yes, this is an excellent point and have added this to the method section.

‘Based on strategies using intense light therapy [10,000 LUX] to treat seasonal mood disorders in humans [35], we adopted a similar protocol. Timing of light therapy has been shown to be critical as light therapy e.g. if given at night can cause insomnia and hyperactivity and therefore the morning hours are recommended.’

Reviewer #2 

Reviewer #2: This article is very interesting and significant. In my opinion the study was planned very well.

Thank you so much.

I have some doubts about small number of human volunteers – too small?

Thank you so much for pointing this out. The human volunteer number for the protein array is clearly limited. This is mainly due to the cost of the array. However, 12 samples from 4 volunteers were enough to reach statistical significance in a large amount of proteins. In addition, an n of 3 per timepoint/condition for arrays is not unusual. Since we have 12 arrays and a time kinetic, we think our data are in fact quite strong.

For the clotting studies in human volunteers we had in fact an n of 6 individuals for each time point which also resulted in statically significant data points.

- number of mice used in experiments - there is no information.

The n numbers are all mentioned in the figure legends in bold. We have used an n of 4 to 6 which has been sufficient to reach statistical significance based in a large experience from in prior studies. 

- tests which were used to perform a statistical analysis - a small number of healthy human volunteers

2 group analysis, which were most of the experiments, were analyzed using a t test. The protein array data were analyzed via linear regression with false discovery rate correction of the p values using the Benjamini and Hochberg method to control for multiple comparisons.

---

## [Editor Report · Decision Letter 1]

17 Dec 2020

Intense light as anticoagulant therapy in humans

PONE-D-20-20729R1

Dear Dr. Eckle,

We’re pleased to inform you that your manuscript has been judged scientifically suitable for publication and will be formally accepted for publication once it meets all outstanding technical requirements.

Kind regards,

Paul A. Bartell

Academic Editor

PLOS ONE
---

## [Editor Report · Acceptance letter]

21 Dec 2020

PONE-D-20-20729R1 

Intense light as anticoagulant therapy in humans 

Dear Dr. Eckle:

I'm pleased to inform you that your manuscript has been deemed suitable for publication in PLOS ONE. Congratulations! Your manuscript is now with our production department. 

Kind regards, 

on behalf of

Dr. Paul A. Bartell 

Academic Editor

PLOS ONE